# Effect of Chilled Storage on Antioxidant Capacities and Volatile Flavors of Synbiotic Yogurt Made with Probiotic Yeast *Saccharomyces boulardii* CNCM I-745 in Combination with Inulin

**DOI:** 10.3390/jof8070713

**Published:** 2022-07-06

**Authors:** Abid Sarwar, Sam Al-Dalali, Tariq Aziz, Zhennai Yang, Jalal Ud Din, Ayaz Ali Khan, Zubaida Daudzai, Quratulain Syed, Rubina Nelofer, Nazif Ullah Qazi, Zhang Jian, Anas S. Dablool

**Affiliations:** 1Beijing Advanced Innovation Center for Food Nutrition and Human Health, Beijing Engineering and Technology Research Center of Food Additives, Beijing Technology and Business University, Beijing 102401, China; abiduomian@gmail.com (A.S.); iwockd@gmail.com (T.A.); jallal4827@gmail.com (J.U.D.); zhangjian@btbu.edu.cn (Z.J.); 2Food & Biotechnology Research Center (FBRC), Pakistan Council of Scientific Industrial Research (PCSIR), Lahore 54600, Pakistan; llcpcsir@gmail.com (Q.S.); rubinanelofer@gmail.com (R.N.); 3School of Food and Biological Engineering, Hefei University of Technology, Hefei 230601, China; salihsam4@gmail.com; 4Pak-Austria Fachhochschule, Institute of Applied Sciences and Technology, Haripur 22621, Pakistan; 5Department of Biotechnology, University of Malakand, Chakdara 18800, Pakistan; ayazkhan@uom.edu.pk; 6Department of Bioresource and Technology, King Mongkut’s University of Technology Thonburi, Bangkok 10140, Thailand; zubaida.khan@mail.kmutt.ac.th; 7Department of Microbiology, University of Swabi, Ambar 94640, Pakistan; nazifullah@gmail.com; 8Department of Public Health, Health Sciences College Al-Leith, Umm Al-Qura University, Makkah al-Mukarramah 24382, Saudi Arabia; asdablool@uqu.edu.sa

**Keywords:** antioxidant, volatiles, synbiotic, *Saccharomyces boulardii*, inulin

## Abstract

Fermentation of available sugars in milk by yogurt starter culture initially and later by *Saccharomyces boulardii* (Probiotic yeast) improves the bioavailability of nutrients and produces bioactive substances and volatile compounds that enhance consumer acceptability. The combination of *S. boulardii*, a unique species of probiotic yeast, and inulin, an exopolysaccharide used as a prebiotic, showed remarkable probiotic and hydrocolloid properties in dairy products. The present study was designed to study the effect of fermentation and storage on antioxidant and volatile capacities of probiotic and synbiotic yogurt by incorporation of *S. boulardii* and inulin at 1%, 1.5%, and 2% (*w*/*v*), compared with the probiotic and control plain yogurt. All samples were stored at 4 °C, and during these four weeks, they were analyzed in terms of their antioxidant and volatile compounds. The synbiotic yogurt samples having inulin and *S. boulardii* displayed significantly higher DPPH (2,2-diphenyl-1-picrylhydrazyl) free radical activity values and more values of TPC (total phenol contents) than control plain yogurt. A total of 16 volatile compounds were identified in S5-syn2 and S4-syn1.5, while S3-syn1 and S2-P had 14, compared with the control S1-C plain yogurt samples, which had only 6. The number of volatile compounds increased with the increasing concentration of inulin throughout the storage period. Therefore, this novel synbiotic yogurt with higher antioxidant and volatile compounds, even with chilling storage conditions, will be a good choice for consumer acceptability.

## 1. Introduction

Functional foods contain one or more specific compounds that have a functional effect on improving the health and well-being of the consumer. These beneficial components can be naturally increased in the food or intentionally added during the production process to produce health effects such as regulating metabolic activities and fitness and improving digestive systems, heart, vessels, etc. [1]. Probiotic products are one of the most common types of functional foods. In recent years, there has been an increasing effort to use probiotic microorganisms to produce various foods in which they have been marketed, including probiotic yogurts, cheese, and fermented beverages [2].

Reactive oxygen species (ROS) are reactive chemical individuals incorporating oxygen atoms with an unpaired electron (radical) or O–O bond and can participate in chemical reactions [3]. ROS have crucial roles in the basic biological processes in the body. These molecules participate in oxidation and reduction reactions (in the respiratory chain) and the removal of toxins, renew energy (ATP), enable oxygen transport by hemoglobin, and activate cytochrome P450 and phagocytosis of microorganisms. However, their overproduction can lead to free radical reactions, resulting in damage to lipids, proteins, carbohydrates, and nucleic acids. The quick and easy diffusion of ROS and their ability to react with multiple nonspecific components of cells may lead to disturbances of many biological processes, resulting in the development of many diseases [4].

Antioxidant compounds in foods play significant roles as health-protecting factors. They can deactivate free radicals, which can cause cell and tissue damage. These types of damage cause malfunctioning of cells or cell death. Epidemiological studies have shown that antioxidants can prevent the development of degenerative diseases such as cancer, coronary heart diseases, obesity, type 2 diabetes, hypertension, premature aging, and inflammatory diseases [5].

Yogurt is the best-known nutritional carrier for the efficient transfer of beneficial microbes into the body [6]. Recently, there has been an intensified demand for a new range of dairy products such as cheese, various lactic acid bacteria drinks, combined probiotic (fermented) milk products, and several types of yogurts, including synbiotic yogurt containing both probiotics and prebiotics [7,8]. Lactic acid bacteria also produce abundant bactericidal proteins in dairy foods [9,10]. Synbiotic yogurt has become increasingly popular as a type of functional food that beneficially affects human health conditions [11,12].

Tibetan kefir grains (TKGs) contain a complex synbiotic diversity of microorganisms, including lactic acid bacteria (LAB), yeast, and acetic acid bacteria. They are a unique natural dairy starter that originated in Tibet, China [13]. The milk is fermented into Tibetan kefir, which has been shown to be an impressive functional food with health benefits including anti-inflammatory effects [14], cholesterol-lowering ability [15], and antioxidant activities [16]. During the fermentation process, yeasts and LAB interact with each other. Yeasts provide vitamins, amino acids, and other essential growth factors for bacteria, and bacterial metabolic end-products are used as energy sources by yeast [17,18].

Native bacteria are not probiotics unless they are isolated, purified, and proven beneficial to health when introduced in animal models in vivo. Prebiotics are a nondigestible part of food, which may serve as nutritional supplements for probiotic microorganisms to enhance their survival chances and implantation in the host intestinal tract [19]. Thus, prebiotics cause particular changes in the composition and activity of the gastrointestinal microflora that benefit the host’s well-being and health [20]. Probiotics may provide a potentially promising approach to preventing microbial dysbiosis [21]. However, synbiotics can better influence lipid profiles and protect against colorectal cancer than probiotics or prebiotics alone [22].

*Saccharomyces boulardii* has been previously identified as a unique species of yeast and characterized as a probiotic strain among a few probiotic yeasts reported to date [23]. Unlike other *Saccharomyces* strains with optimal growth at about 30 °C, *S. boulardii* survives best at 37 °C, which is advantageous, as it is one of the few yeasts with the best performance in the human body temperature [24]. *S. boulardii* is considered a safe microorganism, with nontoxic and nonpathogenic effects. It can be implanted in large quantities in the gastrointestinal tract maintaining a constant level of viability [25]. A bio-therapeutic agent based on the use of *S. boulardii* was developed via oral administration of this probiotic strain to treat recurrent *Clostridium difficile*-associated disease [26].

Inulin, one of the most common prebiotics, is mainly found in roots of chicory (*Cichorium intybus*), garlic (*Allium sativum*), wheat (*Triticum* spp.), oat (*Avea sativa*), and Dalia (*Bulgar*) [27]. It is known to be a storage polymer consisting of a *β*-2-1-linked fructosyl unit with a terminal glucosyl unit [28]. At present, there is an increasing interest in the addition of inulin and other oligofructoses to food products (e.g., yogurt) for their healthful effects, e.g., enhancing *Lactobacillus* and *Bifidobacterium* growth in the colon, boosting the bioavailability of a variety of minerals such as calcium and iron, increasing antioxidant activities, and boosting immune functions [29]. Inulin and oligofructose improved sensory quality and increased the viable probiotic count in functional dairy foods [30]. Supplementation of food fibers such as inulin could reduce wheying-off, thus improving the textural properties of the food matrix, and it was also found to remarkably elevate viscosity and shear thinning behavior of different dairy products [31]. 

Previously, different yogurt samples containing this probiotic yeast with different inulin concentrations showed the best physicochemical, microbiological, sensory properties, microrheology, and microstructure [32]. Owing to the interest of consumers and increasing demands of the food industry, in the present study, synbiotic yogurt having probiotic *S. boulardii* and prebiotic inulin in different concentrations were evaluated in terms of their antioxidants and volatile compounds during 28 days of storage. 

## 2. Materials and Methods

### 2.1. Microorganisms and Culture Condition

Probiotic yeast (*S. boulardii*) was purchased as a lyophilized powder in the form of a sachet (Martin Dow). The yeast culture (8.22 ± 0.28 CFU/mL) of *S. boulardii* was prepared according to the method described by Eunice et al. (2017) and used in yogurt making. The number of colony-forming units of *S. boulardii* (CFU/g) was determined using Sabouraud dextrose agar in different dilutions made by dissolving a 250 mg sachet in 9 mL peptone water [33]. The yoghurt starter culture containing *L. delbrueckii* ssp. *bulgaricus* and *Streptococcus thermophilus* was purchased in powder form (DANISCO, France) and activated by consecutively transferring it three times to 10% (*w*/*v*) reconstituted skim milk at 37 °C for 24 h. Inulin was purchased in powder form (Digestive-Now).

### 2.2. Preparation of Yogurt

High-quality fresh cow milk was purchased from a local dairy (Beijing SanYuan-Dairy Co., Ltd., Beijing, China). Five experimental groups of yogurt were arranged (Table 1)—namely, control plain yogurt (S1-C), probiotic yogurt with 0.5% *S. boulardii* (S2-P), synbiotic yogurt with 0.5% *S. boulardii* + 1% inulin (S3-Syn1), 0.5% *S. boulardii* + 1.5% inulin (S4-Syn1.5), and 0.5% *S. boulardii* + 2% inulin (S5-Syn2)—which were made according to the method described by [32,33], with some modifications. After blending the fresh milk with 5% skim milk powder, the mixture was homogenized (performed with a homogenizer), pasteurized (85 °C, 30 min), and cooled to 43 °C. The mixture was inoculated with 3% (*w*/*v*) of the yogurt starter culture and mixed well. Then, the *S. boulardii* culture and inulin were added, as described in Table 1. The yogurt samples were incubated at 43 °C ± 2 °C until reaching approximately pH 4.5, and then they were stored at 4 °C for four weeks. Sampling was performed every week during the storage for subsequent analyses.

### 2.3. Antioxidant Activities of Yogurt Samples

The antioxidant activities (DPPH, 2,2-diphenyl-1-picrylhydrazyl, and TPC, as well as total phenol contents) were determined on different days of cold storage on days 0, 7, 14, 21, and 28. 

#### 2.3.1. DPPH (2,2-Diphenyl-1-Picrylhydrazyl) Free Radical Scavenging Assay

The antioxidant activity of the yogurt samples was determined using DPPH free radical scavenging assay according to [34], with some modifications. Briefly, a 1/10 (*w*/*v*) dilution of the yogurt sample in water was prepared. Then, an aliquot of 0.1 mL of each sample was added to 4 mL of DPPH solution with a concentration of 0.004% in methanol. The mixture was shaken thoroughly and allowed to stand at room temperature for 30 min. The OD of the samples was read at 517 nm against water as blank. The inhibition percent (I%) of the DPPH free radical was calculated. The readings were recorded in triplicates.

The percentage of radical scavenging activity (RSA) was calculated as follows:(1)$$RSA\;\left(\%\right)=\frac{A\;\left(control\right)-A\;\left(sample\right)}{A\;\left(control\right)}\times 100$$

#### 2.3.2. Total Phenol Content Determination

The total phenolic content of yogurt samples was measured using the Folin–Ciocalteu method. Briefly, 5 g of yogurt samples were mixed with 15 mL distilled water in centrifuge tubes and centrifuged at 4000 g for 10 min. A 0.1 mL aliquot of supernatant of each sample was mixed with 4 mL of 50% (*v*/*v*) Folin–Ciocalteu reagent, and 2 mL of 2% sodium carbonate solution was added to the mixture and allowed to stand for 2 h. The OD of the samples was read at 750 nm against a solution consisting of 4 mL Folin–Ciocalteu and 2 mL sodium carbonate as blank. Gallic acid was used as a standard to prepare the calibration curve. The total phenolic content of the yogurt samples is expressed as mg gallic acid/kg yogurt [35].

### 2.4. Volatile Analysis 

#### 2.4.1. Extraction of Volatile Flavor Compounds

The volatile compounds in yogurt samples were analyzed using headspace solid-phase microextraction–gas chromatography–mass spectrometry (HS-SPME–GC–MS) on days 1, 7, 14, 21, and 28 of chilled storage. Briefly, a 5 g sample was mixed with 2 µL of 1,2-dichlorobenzene as internal standard (I.S, 1500 ppm) in a 20 mL headspace vial, which was then firmly closed with a silicon septum. The vial was placed in a water bath for 60 min to equilibrate at 55 °C, after which the volatile compounds were extracted using a fiber manufactured from StableFlex divinylbenzene carboxen polydimethylsiloxane (DVB-CAR-PDMS) in 50/30 μm (Supelco, Bellefonte, PA, USA) at the same temperature. The absorbed volatile compounds were then desorbed in the injection port of a GC–MS for 5 min in a splitless mode at 250 °C.

#### 2.4.2. GC–MS Analysis

A polar column (DB-WAX, 30 m 0.25 mm 0.25 m, Agilent Technology, Santa Clara, CA, USA) was used in a PerkinElmer GC–MS (GC, Clarus 680; MS, Model SQ8C) using helium as the carrier gas at a follow rate of 1 mL/min. The initial oven temperature was held at 35 °C for 5 min, then ramped up to 140 °C at 4 °C/min and kept for 5 min, before being elevated to 230 °C at 10 °C/min and kept for 5 min. An MS detector was used with a mass scan range of 33–450 amu (*m*/*z*) and a 70 eV electron ionization (EI) voltage. Both the ion source and line transfer were adjusted at 250 °C.

#### 2.4.3. Identification and Quantitation of Volatile Flavor Compounds

The retention indices (RIs) and mass spectra (MS) of volatiles in the DB-WAX column were compared with those published in the National Institute of Standards and Technology library (NIST 14).

The internal standard method was used for quantitation analysis, in which the peak area of the detected volatiles was compared with the internal standard’s matching peak area. The following equation was used to calculate the concentration of each compound: $$Con.\;\left(\mathrm{mg}/100\;g\right)=\frac{Peak\;area\;ratio\;\left(\frac{volatile}{internal\;standard}\right)\times \mathrm{con}.\;\mathrm{of}\;internal\;standard}{Sample\;weight}\times 100$$

### 2.5. Statistical Analysis

All measurements were performed in triplicate. The results were statistically analyzed with one-way ANOVA using Statistix 8.1 software. The volatile compounds data were also subjected to multivariate statistical techniques in terms of principal component analysis (PCA) and hierarchical cluster analysis (HCA).

## 3. Results and Discussion

### 3.1. Antioxidant Activities of Yogurt Samples

The antioxidant activities (DPPH and TPC) were determined in the analyzed yogurt samples throughout the storage period. 

#### 3.1.1. Radical Scavenging Assay during Storage 

The DPPH radical is one of the rare stable organic nitrogen radicals and can take up an electron or hydrogen radical to become a stable diamagnetic molecule. The stable DPPH radical scavenging model is a broadly used method to evaluate antioxidant activity in a short period of time, compared with other methods. The effect of antioxidants on DPPH radical scavenging is related to their ability to donate hydrogen [36]. The DPPH activities of probiotic and synbiotic yogurt samples were lowest after day 1 storage, when values, including control (S1-C), were below 50%. However, after 14 days of storage, the activities of probiotic (S2-P = 54.65%) and synbiotic (S3-Syn1 = 61.01%, S4-Syn1.5 = 62.73%, and S5-Syn2 = 65.51%) yogurt samples were significantly increased, compared with that of the control sample (S1-C = 26.44%), as shown in Table 2. Increased antioxidant activity in probiotic and synbiotic yogurt may result from bioactive (antioxidative) peptides released from protein digestion via bacterial fermentation. Similarly, the pH of yogurt, because of the fermentation of available lactose by yogurt culture, becomes acidic, which creates a favorable environment for *S. boulardii* to utilize the remaining sugars available and produce more bioactive compounds. 

All the yogurt samples displayed radical scavenging capacities in descending order, i.e., S5-Syn2 < S4-Syn1.5 < S3-Syn1 < S2-P < S1-C. The addition of inulin in synbiotic yogurt also contributed to the higher antioxidant activities. In previous studies, Valentina et al. [37] reported that inulin had significantly higher antioxidant activity than other sugars, even after cooking and digestion. In another study, Jansen et al. [38] observed that probiotics in yogurt drinks stored at refrigerated temperature had higher antioxidant activity than those in yogurt drinks stored at room temperature. DPPH activity was increased in yogurt with the addition of an aqueous extract of *Matricaria recutita* [39]. Hydrolysis of milk protein or organic acid production can be another reason for yogurt’s antioxidant activity because of microbial metabolic activity during fermentation and storage under refrigeration [40].

#### 3.1.2. Total Phenol Content of Yogurt Samples during Storage

The total phenol content in all yogurt samples was significantly (*p* ≤ 0.05) different during the refrigerated storage period. All synbiotic samples showed higher phenol content with respect to the concentration of inulin used, as presented in Table 3. Among the samples, S5-Syn2 showed the best results, at 4.02 mg.GAE/g, while S1-C had the lowest, at 0.57 mg.GAE/g. A higher concentration of TPC up to 6.30 mg GAE/g on the 28th day of storage was reported by Adriana et al. (2018) [41] in yogurt supplemented with Marjoram (*Origanum vulgare*) extract. All synbiotic yogurt samples (S3-Syn1 = 3.54 mg.GAE/g, S4-Syn1.5 = 3.75 mg.GAE/g, and S5-Syn2 = 4.02 mg.GAE/g) supplemented with 1%, 1.5%, and 2% inulin were also rich in total phenol content at the end of cold storage (28th day), compared with probiotic (S2-P = 1.52 mg.GAE/g) and control plain yogurt (S1-C = 0.78 mg.GAE/g).

The higher TPC in synbiotic yogurt is due to inulin, a fructan polysaccharide found in many different types of plants, such as chicory, artichoke, salsify, and Jerusalem artichoke. Senadeera et al. (2018) [42] observed the highest phenolic content of 15.53 ± 0.46 mg GAE/100 g in yogurt fortified with Annona species pulp. In another study, Chouchouli et al. [43] observed a higher TPC value for yogurt containing grape seed extracts. Moreover, the addition of strawberry pulp enhanced the phenolic content and antioxidant properties of yogurt [44]. Furthermore, green tea supplementation increased the TPC value of probiotic yogurt during refrigerated storage [45]. In a recent study, a higher TPC value was observed in yogurt supplemented with osmo-air-dried mulberry, compared to the control [46].

### 3.2. Volatile Compounds of Synbiotic Yogurt

The volatile compounds of synbiotic, probiotic, and control yogurt were determined using GC–MS during cold storage for up to four weeks. Flavor in yogurt is formed by the action of yogurt starter bacteria and originated from biochemical changes in carbohydrates, lipids, and proteins. The flavor is frequently the first indicator when consumers choose food. Consumers will not be interested in functional food consumption if biologically active ingredients lead to unpleasant flavor [47]. The volatile analysis is widely applied in the objective assessment of dairy foods, to determine consumers’ acceptance of new functional products [48]. 

Storage conditions greatly influenced the formation of volatile compounds in the synbiotic yogurt made with probiotic *S. boulardii* and inulin compared with the control plain yogurt and the probiotic yogurt sample using yeast without inulin, as shown in Table 4. Analysis with HS-SPME–GC–MS showed that a total of 14 volatile compounds were identified in the probiotic yogurt (S2-P) in week 0, 13 in weeks 1 and 2, 10 in week 3, and 8 in week 4, compared with control (S1-C), having only 5 in week 0 and 6 in week 1, which was reduced to 4 in the refrigerated reaming storage. However, synbiotic yogurt samples showed a higher concentration of volatile compounds throughout the storage period. A total of 14 volatiles were identified in S3-Syn1 in week 0, 14 in week 1, 13 in week 2, 12 in week 3, and 11 in week 4. Similarly, in synbiotic samples (S4-Syn1.5, S5-Syn2), 16 volatiles were formed that were reduced to 13 and 12, respectively, in week 4 of cold storage. 

These volatile compounds were from different chemical families, but limonene and *L*-limonene (Terpene family) were present in probiotic and synbiotic yogurt samples having *S. boulardii* in common. Among these samples, S3-Syn1 had the highest limonene concentration (162.28 ± 7.30 mg/100 g), while S5-Syn2 had more *L*-limonene concentration (102.29 ± 5.66 mg/100 g) in weeks 4 and 2, respectively. Limonene is generally considered safe as a food additive or flavoring and a fragrance additive. Previous studies in the literature have revealed that limonene can present antimicrobial, antifungal, antimalarial, and antitumoral activities [49]. 

A total of six volatile compounds in the control S1-C were formed—namely, one alcohol (ethanol), one ketone (acetoin), and four acids (acetic acid, butanoic acid, hexanoic acid, and octanoic acid) at the end of storage. In the fourth week, only four volatiles (ethanol = 1.72 ± 0.66 mg/100 g, acetoin = 36.16 ± 3.09 mg/100 g, hexanoic acid = 9.60 ± 0.91 mg/100 g, and octanoic acid = 14.37 ± 1.40) were present. 

Among alcohols, ethanol was the predominant compound present in all yogurt samples, but the concentration was lower in S1-C, ranging from 0.83 ± 0.15 at week 1 to 1.72 ± 0.66 mg/100 g at week 4, compared with probiotic S2-P, ranging from 28.13 ± 2.76 at week 1 to 59.89 ± 4.33 mg/100 g at week 4, and much higher concentration in synbiotic yogurt samples S3-Syn1, ranging from 93.32 ± 7.34 at week 1 to 306.81 ± 22.08 mg/100 g at week 4 of the refrigerated storage period. These results indicated that it is a low alcohol product, compared with kefir fermented with yeast and lactic acid bacteria. In terms of the final fermentation products, kefir lactic acid and ethanol were the main ones, whereas only traces of acetic acid were detected. The concentration of ethanol was ~0.9% *w*/*v* (~9 g/L), as reported by Tzavaras et al. [50]. Similarly, Chen et al. [51] also reported that ethanol was one of the main components in volatiles, in their study of the effect of lactic acid bacteria and yeasts on the structure and fermentation properties of Tibetan kefir grains. The other alcohol (phenylethyl alcohol) was only found in the synbiotic yogurt samples (S4-Syn1.5 = 13.09 ± 0.98 and S5-Syn2 = 16.51 ± 2.11 mg/100 g at week 3), having a higher concentration of inulin. Yeast cells are capable of producing phenyl ethyl alcohols via normal metabolism. This can be synthesized through two independent routes in yeast cells, either de novo via the Shikimate pathway or via the Ehrlich pathway [52]. Alcoholic compounds also contribute to flavor improvement in fermented dairy products. 

A total of four acids (acetic acid, butanoic acid, hexanoic acid, and octanoic acid) were identified in the yogurt samples having higher concentrations in the descending order of acetic acid ≤ hexanoic acid ≤ octanoic acid ≤ butanoic acid found in synbiotic samples, followed by probiotic and control. Acetic acid was one of the main acids in yogurt samples, and significant differences were observed among synbiotic, probiotic, and control samples in terms of its concentration. Acidity is a decisive factor for flavor acceptance of yogurt and is maintained near pH 4.5 [53]. Acid compounds are generally present in various fermented dairy products [54].

Two short-chain fatty acid compounds (propanoic acid and 2-methyl-butanoic acid) were present in probiotic and synbiotic samples having *S. boulardii* in common. However, they were not found in control plain yogurt. This might be credited to the ability of *S. boulardii* CNCM I-745 to secrete alkaline phosphatase, which causes dephosphorylation of phospholipid substrates to generate phosphate and other short-chain fatty acids [55,56]. 

Three esters (1-butanol-3-methyl-acetate, butanoic acid-3-methyl butyl ester, and octanoic acid ethyl ester) were identified in only probiotic and synbiotic yogurt samples. 1-Butanol-3-methyl-acetate ester was detected in all of the probiotic and synbiotic yogurt samples of this study up to four weeks of storage, but butanoic acid-3-methyl butyl ester was detected only up to the second week, while octanoic acid ethyl ester was only found in S4-Syn1.5 and S5-Syn2 having higher concentrations of inulin. Ester volatile compounds are generally produced at low concentrations in dairy products when lactose is fermented with lactic acid bacteria (LAB) [57].

Aldehydes are considered important aroma compounds contributing to the volatile profile of fermented dairy products with lactic acid bacteria [58]. The only aldehyde (3-hydroxy butanal) found was in probiotic yogurt (S2-P = 0.63 ± 0.11 mg/100 g) and synbiotic yogurt (S3-Syn1 = 13.95 ± 3.31 mg/100 g) samples of this study up to one week of storage. The formation of more volatile compounds in the synbiotic yogurt samples enriched its flavor, compared with the plain and probiotic yeast yogurts that contained fewer volatile compounds. Dan et al. (2017) [59] also showed that the aroma profile of yogurt made with pure culture was different from those made with the addition of probiotics alone or with a combination of prebiotics. 

### 3.3. Principal Component Analysis of Volatile Compounds

Volatile compounds identified via GC–MS among the different probiotic, synbiotic, and control yogurt samples (S1-C: control plain yogurt; S2-P: yogurt with *S. boulardii*; S3-Syn1: yogurt with *S. boulardii* + 1% inulin; S4-Syn1.5: yogurt with *S. boulardii* + 1.5% inulin; and S5-Syn2: yogurt with *S. boulardii* + 2% inulin) during four weeks of storage were further analyzed via principal component analysis (PCA) and hierarchical cluster analysis (HCA). Both these multivariate statistical techniques were used for easy interpretation of the volatile compounds’ data from GC–MS. Hierarchical cluster analysis is shown in Figure 1, in which the yogurt samples are divided into three groups, determined by the average linkage among different groups based on Euclidian distance. The principal component analysis explained a total of 96.7% variance, to which principal component 1 (PC1) contributed 90.3%, whereas principal component 2 (PC2) contributed 6.4%, respectively.

Two principal components were used in most cases of PCA analysis, appropriately explaining a great proportion of variations in original parameters. A bi-plot of different yogurt sample scores is shown in Figure 2**,** revealing the most important loadings and the percentage accounted for by the two first principal components (PC1 and PC2) after analysis. There were three different groups, and the samples were positioned based on the concentrations of volatiles and storage weeks. All yogurt samples were positioned on the left side of PC1, but group1 (S1-C and S2-P) were in the negative part of the bi-plot in week 0. The control samples remained in the negative zone throughout the four weeks of storage, having very few volatiles produced. Group 2 samples having probiotic and synbiotic yogurt samples were positioned on the positive side of PC1, especially after 1 week of storage. Samples in group 3 (synbiotic only having both *S. boulardii* and inulin) were positioned on the positive side of PC1 and located together in the bi-plot. Significant differences were observed among old yogurts of these groups at weeks 0 and 4. In summary, the bi-plot indicates that limonene, L-limonene, and carboxylic acids (octanoic acids, hexanoic acid, butanoic acid, and acetic acid), along with ethanol, were grouped on the positive side of PC1, strongly contributing to the flavor of probiotic and synbiotic yogurt samples. Similarly, esters (1-butanol-3-methyl-acetate, butanoic acid-3-methyl butyl ester, octanoic acid ethyl ester) were secondary in contributing to the flavor of probiotic and synbiotic yogurt samples and were also placed on the positive side of PC1. The remaining volatile compounds (propanoic acid and 2-methyl-butanoic acid, 3-hydroxy butanal) were grouped in the center of the bi-plot, with less contribution to the principal component, compared with the other compounds. According to these results, PCA, along with HCA, is a powerful tool to distinguish the samples based on treatment (probiotics, synbiotics) and magnitude of storage time.

## 4. Conclusions

Functional yogurt samples having *S. boulardii* and inulin in different concentrations showed that the antioxidant potential increased during the storage period and remained stable. Similarly, the probiotic and synbiotic yogurt samples showed that the number of volatile compounds also increased with chilled storage. The higher amounts of ethanol (S3-Syn1 = 306.81 ± 22.08 and S4-Syn1.5 = 304.44 ± 18.20 mg/100 g) indicated that *S. boulardii* could ferment the available sugars to produce more volatile compounds, such as limonene, acids, esters, and alcohols, in higher concentrations, while these were either absent or produced in very low concentrations in the control yogurt sample throughout the storage period. Thus, synbiotic yogurt with more antioxidant potential and higher concentrations of volatiles can strengthen consumer acceptability, representing a novel synbiotic dairy product with probiotic yeast.

## Figures and Tables

**Figure 1 jof-08-00713-f001:**
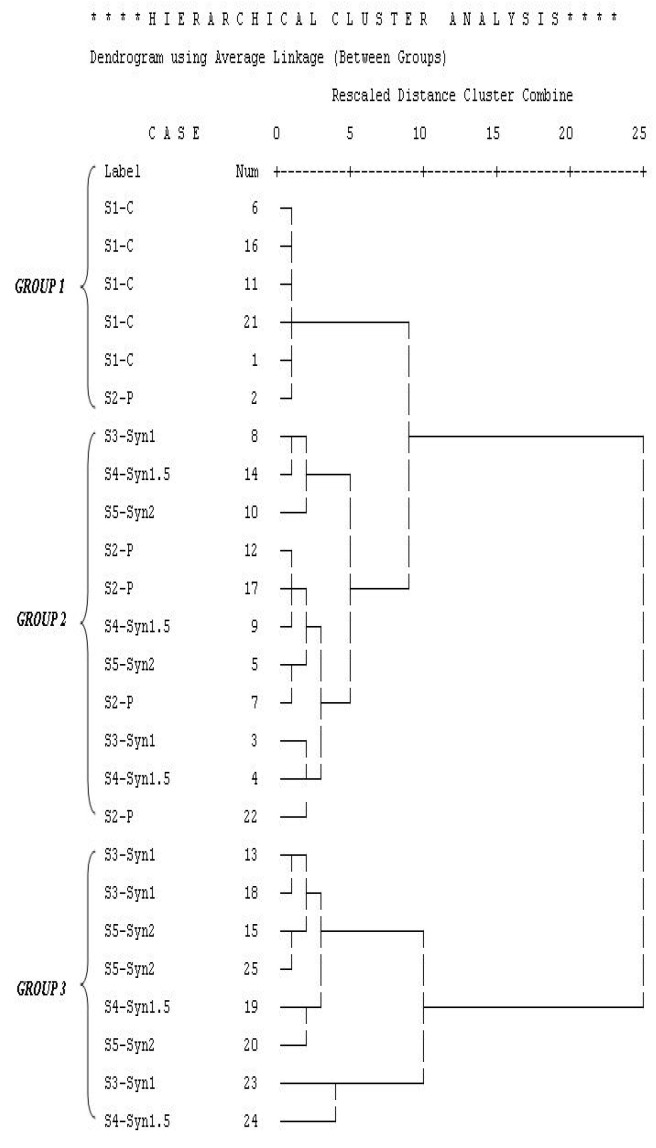
Hierarchical cluster analysis of different yogurt samples on the basis of volatiles produced during storage of four weeks.

**Figure 2 jof-08-00713-f002:**
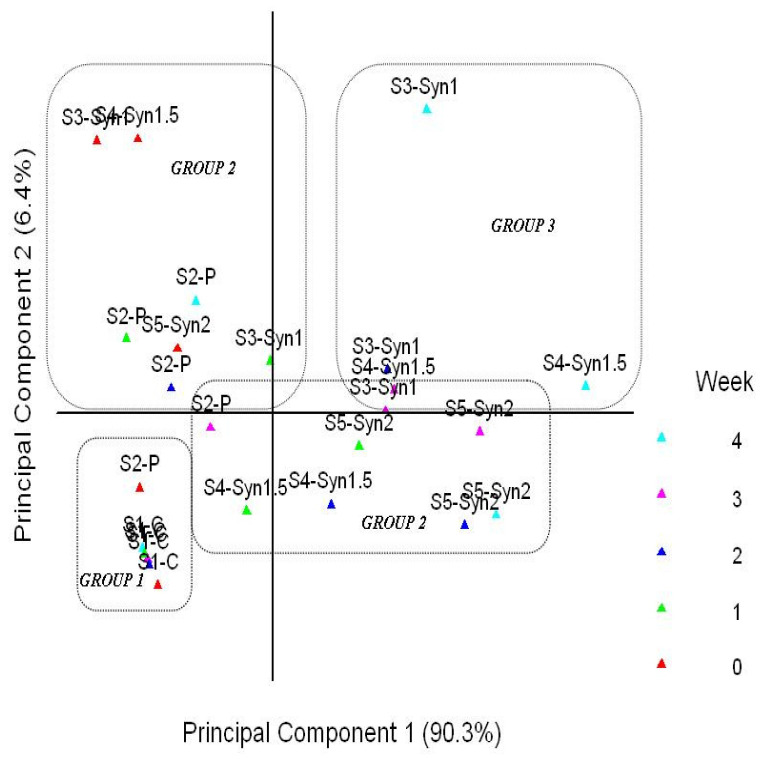
Principal component analysis of different yogurt samples on the basis of volatiles produced during storage of four weeks.

**Table 1 jof-08-00713-t001:** Experimental groups of yogurt samples.

Sample Name	Concentration of *S. boulardii* and Inulin
S1-C	Control plain yogurt
S2-P	Yogurt with 0.5% *S. boulardii*
S3-Syn1	Yogurt with 0.5% *S. boulardii* + 1% inulin
S4-Syn1.5	Yogurt with 0.5% *S. boulardii* + 1.5% inulin
S5-Syn2	Yogurt with 0.5% *S. boulardii* + 2% inulin

**Table 2 jof-08-00713-t002:** Influence of storage time on the antioxidant activity of different yogurt samples.

Name of Sample	Storage Day				
	**1**	**7**	**14**	**21**	**28**
** *S1-C* **	11.65 ± 0.7 ^E^	20.38 ± 0.36 ^E^	26.44 ± 0.8 ^E^	38.74 ± 0.25 ^E^	45.48 ± 0.57 ^E^
** *S2-P* **	11.94 ± 0.54 ^D^	22.44 ± 0.5 ^D^	54.65 ± 0.85 ^D^	64.57 ± 0.43 ^D^	72.32 ± 0.76 ^A^
** *S3-Syn1* **	12.07 ± 0.32 ^C^	27.74 ± 0.86 ^C^	61.01 ± 0.28 ^C^	70.75 ± 0.36 ^c^	70.23 ± 0.65 ^D^
** *S4-Syn1.5* **	14.09 ± 0.87 ^B^	29.50 ± 0.4 ^B^	62.73 ± 0.92 ^B^	71.26 ± 0.84 ^B^	72.03 ± 0.54 ^BC^
** *S5-Syn2* **	15.84 ± 0.48 ^A^	32.80 ± 0.74 ^A^	65.51 ± 0.74 ^A^	72.75 ± 0.27 ^A^	72.14 ± 0.84 ^BC^

Different superscripted letters indicate significant differences (*p* ≤ 0.05) within a column.

**Table 3 jof-08-00713-t003:** Total phenol content of different yogurt samples in mg.GAE/g.

Name of Sample	Storage Day				
	**1**	**7**	**14**	**21**	**28**
** *S1-C* **	0.57 ± 0.07 ^e^	0.62 ± 0.05 ^e^	0.67 ± 0.1 ^e^	0.74 ± 0.09 ^e^	0.78 ± 0.04 ^e^
** *S2-P* **	1.04 ± 0.03 ^d^	1.42 ± 0.07 ^d^	1.65 ± 0.05 ^d^	1.57 ± 0.04 ^d^	1.52 ± 0.06 ^d^
** *S3-Syn1* **	1.74 ± 0.06 ^c^	2.34 ± 0.1 ^c^	2.86 ± 0.08 ^c^	3.21 ± 0.06 ^c^	3.54 ± 0.65 ^c^
** *S4-Syn1.5* **	2.09 ± 0.07 ^b^	2.82 ± 0.04 ^b^	3.13 ± 0.02 ^b^	3.57 ± 0.07 ^b^	3.75 ± 0.09 ^b^
** *S5-Syn2* **	2.76 ± 0.08 ^a^	3.25 ± 0.05 ^a^	3.49 ± 0.09 ^a^	3.75 ± 0.10 ^a^	4.02 ± 0.04 ^a^

Different superscripted letters indicate significant differences (*p* ≤ 0.05) within a column.

**Table 4 jof-08-00713-t004:** Identification and quantitation of flavor-contributing volatile compounds in different yogurt samples during different times of storage.

				Concentration (mg/100 g)									
Compounds	CAS	RI ^a^	Identification	Week 0					Week 1				
				S1-C	S2-P	S3-Syn1	S4-Syn1.5	S5-Syn2	S1-C	S2-P	S3-Syn1	S4-Syn1.5	S5-Syn2
**Ethanol**	64-17-5	932	RI, MS	n.d	10.56 ± 0.61	54.91 ± 11.43	83.97 ± 9.04	64.19 ± 4.54	0.83 ± 0.15	28.13 ± 2.76	93.32 ± 7.34	68.07 ± 3.55	145.24 ± 13.43
**3-Hydroxy butanal**	107-89-1	1009	MS	n.d	0.63 ± 0.11	13.95 ± 3.31	n.d	n.d	n.d	n.d	4.71	n.d	n.d
**2-Methyl-1-propanol**	78-83-1	1092	RI, MS	n.d	2.45 ± 0.21	13.25 ± 1.74	11.56 ± 1.43	7.42 ± 1.32	n.d	4.98 ± 0.79	22.80 ± 1.33	9.75 ± 2.23	21.45 ± 2.77
**1-Butanol-3-methyl-acetate**	123-92-2	1122	RI, MS	n.d	2.84 ± 0.19	8.11 ± 1.13	7.44 ± 0.86	4.90 ± 0.65	n.d	4.14 ± 1.13	6.18 ± 1.21	1.96 ± 0.54	4.79 ± 1.14
**Limonene**	138-86-3	1200	RI, MS	n.d	23.03 ± 1.45	148.99 ± 17.56	136.99 ± 12.13	46.78 ± 5.16	n.d	54.36 ± 6.32	56.05 ± 4.32	20.95 ± 2.44	20.82 ± 2.12
** *L* ** **-Limonene**	5989-54-8	1204	RI, MS	n.d	24.83 ± 3.27	2.42 ± 0.69	12.22 ± 2.14	16.69 ± 2.32	n.d	22.78 ± 2.77	47.64 ± 4.13	31.65 ± 2.69	86.64 ± 4.65
**3-Methyl-1-butanol,**	123-51-3	1209	RI, MS	n.d	31.85 ± 4.12	195.81 ± 12.32	252.81 ± 20.21	173.12 ± 11.35	n.d	108.18 ± 8.19	308.57 ± 20.26	172.47 ± 9.66	370.68 ± 17.48
**Butanoic acid-3-methyl butyl ester**	106-27-4	1259	RI, MS	n.d	2.13 ± 0.76	6.78 ± 1.55	6.12 ± 1.13	4.59 ± 0.84	n.d	6.41 ± 1.54	5.73 ± 0.91	3.02 ± 0.68	5.24 ± 0.89
**Acetoin**	513-86-0	1284	RI, MS	9.46 ± 1.47	62.82 ± 8.37	179.17 ± 9.87	161.69 ± 5.87	108.11 ± 10.21	25.96 ± 3.56	125.46 ± 6.12	117.05 ± 11.43	48.20 ± 3.41	100.22 ± 7.88
**Octanoic acid ethyl ester**	106-32-1	1435	RI, MS	n.d	n.d	n.d	8.81 ± 0.78	6.08 ± 0.93	n.d	n.d	n.d	4.88 ± 0.89	9.70 ± 1.47
**Acetic acid**	64-19-7	1449	RI, MS	1.18 ± 0.35	10.38 ± 1.35	33.45 ± 2.68	65.91 ± 8.32	34.56 ± 6.32	2.71 ± 0.57	30.76 ± 2.28	44.83 ± 2.76	18.78 ± 3.32	42.68 ± 3.68
**2-Methyl-propanoic acid**	79-31-2	1570	RI, MS	n.d	5.51 ± 0.93	25.43 ± 2.88	45.36 ± 3.68	33.46 ± 4.35	n.d	17.97 ± 1.61	27.68 ± 3.23	13.04 ± 1.77	24.30 ± 1.94
**Butanoic acid**	107-92-6	1625	RI, MS	0.54 ± 0.12	6.42 ± 0.84	24.63 ± 4.12	35.15 ± 2.54	23.94 ± 3.45	3.88 ± 0.84	19.06 ± 2.44	18.07 ± 0.89	7.02 ± 0.86	15.32 ± 2.28
**2-Methyl-butanoic acid**	116-53-0	1662	RI, MS	n.d	n.d	n.d	18.36 ± 2.55	12.83 ± 1.57	n.d	n.d	n.d	4.37 ± 0.68	n.d
**Hexanoic acid**	142-62-1	1846	RI, MS	1.35 ± 0.21	21.31 ± 2.75	73.86 ± 7.28	107.78 ± 10.43	61.86 ± 5.61	12.54 ± 1.73	52.35 ± 3.69	44.02 ± 2.43	16.38 ± 2.44	45.00 ± 7.12
**Phenylethyl alcohol**	60-12-8	1906	RI, MS	n.d	n.d	n.d	15.16 ± 1.48	10.06 ± 1.48	n.d	n.d	n.d	5.33 ± 0.88	12.54 ± 1.43
**Octanoic acid**	124-07-2	2060	RI, MS	1.01 ± 0.11	13.93 ± 1.87	44.11 ± 3.69	79.93 ± 4.78	75.08 ± 4.12	20.70 ± 3.27	47.25 ± 5.12	32.51 ± 2.55	17.39 ± 2.41	34.45 ± 3.22
				**Concentration (mg/100 g)**									
**Compounds**	**CAS**	**RI ^a^**	**Identification**	**2 Weeks**					**3 Weeks**				
				**S1-C**	**S2-P**	**S3-Syn1**	**S4-Syn1.5**	**S5-Syn2**	**S1-C**	**S2-P**	**S3-Syn1**	**S4-Syn1.5**	**S5-Syn2**
**Ethanol**	64-17-5	932	RI, MS	0.65 ± 0.11	37.97 ± 3.43	159.72 ± 20.22	120.71 ± 17.11	201.49 ± 21.44	1.79 ± 0.57	44.52 ± 3.09	151.46 ± 9.67	237.85 ± 18.22	289.47 ± 10.55
**3-Hydroxy butanal**	107-89-1	1009	MS	n.d	n.d	25.43 ± 2.14	n.d	n.d	n.d	n.d	24.66 ± 1.77	n.d	n.d
**2-Methyl-1-propanol**	78-83-1	1092	RI, MS	n.d	7.00 ± 0.89	n.d	18.82 ± 1.88	33.54 ± 6.12	n.d	9.33 ± 0.89	n.d	59.82 ± 6.31	39.88 ± 3.65
**1-Butanol-3-methyl-acetate**	123-92-2	1122	RI, MS	n.d	4.61 ± 0.78	10.76 ± 1.12	1.70 ± 0.23	2.06 ± 0.45	n.d	4.94 ± 0.66	8.63 ± 0.88	5.09 ± 0.87	6.79 ± 1.33
**Limonene**	138-86-3	1200	RI, MS	n.d	64.01 ± 5.61	55.49 ± 3.58	13.04 ± 1.06	n.d	n.d	59.03 ± 4.22	45.57 ± 2.68	59.65 ± 4.22	43.95 ± 3.20
**L-Limonene**	5989-54-8	1204	RI, MS	n.d	50.67 ± 5.88	68.25 ± 3.88	41.77 ± 2.43	102.29 ± 5.66	n.d	60.45 ± 3.77	51.89 ± 5.12	6.33 ± 1.11	54.18 ± 2.63
**3-Methyl-1-butanol,**	123-51-3	1209	RI, MS	n.d	143.33 ± 8.38	475.24 ± 25.67	296.07 ± 14.06	468.13 ± 25.77	n.d	179.60 ± 10.37	446.59 ± 28.67	435.72 ± 22.06	526.29 ± 25.12
**Butanoic acid-3-methyl butyl ester**	106-27-4	1259	RI, MS	n.d	3.29 ± 0.65	8.40 ± 0.96	4.33 ± 0.79	n.d	n.d	n.d	n.d	n.d	n.d
**Acetoin**	513-86-0	1284	RI, MS	24.30 ± 3.43	103.86 ± 12.11	118.22 ± 13.59	63.48 ± 3.52	85.09 ± 7.22	22.26 ± 2.12	79.31 ± 7.12	90.96 ± 7.38	100.52 ± 8.19	100.24 ± 6.22
**Octanoic acid ethyl ester**	106-32-1	1435	RI, MS	n.d	n.d	n.d	9.94 ± 0.95	14.03 ± 2.33	n.d	n.d	n.d	18.00 ± 2.55	18.91 ± 1.05
**Acetic acid**	64-19-7	1449	RI, MS	n.d	21.23 ± 4.11	45.19 ± 3.44	31.56 ± 2.66	35.31 ± 3.66	n.d	27.37 ± 2.12	47.26 ± 3.23	70.57 ± 4.22	63.58 ± 2.99
**2-Methyl-propanoic acid**	79-31-2	1570	RI, MS	n.d	9.28 ± 1.23	22.77 ± 1.87	17.22 ± 1.42	15.38 ± 2.43	n.d	n.d	15.46 ± 1.61	17.26 ± 1.05	11.86 ± 0.83
**Butanoic acid**	107-92-6	1625	RI, MS	n.d	11.77 ± 1.12	13.13 ± 1.56	7.29 ± 0.82	4.22 ± 0.88	n.d	n.d	13.02 ± 0.93	11.80 ± 1.11	9.48 ± 0.88
**2-Methyl-butanoic acid**	116-53-0	1662	RI, MS	n.d	n.d	n.d	n.d	n.d	n.d	n.d	n.d	n.d	n.d
**Hexanoic acid**	142-62-1	1846	RI, MS	6.36 ± 1.12	32.74 ± 2.54	37.88 ± 4.55	18.28 ± 1.33	17.88 ± 1.79	18.22 ± 1.54	31.05 ± 1.42	33.84 ± 3.23	31.62 ± 2.44	34.13 ± 3.12
**Phenylethyl alcohol**	60-12-8	1906	RI, MS	n.d	n.d	n.d	4.93 ± 0.58	10.65 ± 1.33	n.d	n.d	n.d	13.09 ± 0.98	16.51 ± 2.11
**Octanoic acid**	124-07-2	2060	RI, MS	9.17 ± 1.67	17.99 ± 1.08	29.80 ± 3.12	16.87 ± 2.22	24.82 ± 3.54	15.53 ± 1.44	21.59 ± 1.68	27.29 ± 3.11	26.65 ± 0.79	29.06 ± 2.07
				**Concentration (mg/100 g)**									
**Compounds**	**CAS**	**RI ^a^**	**Identification**	**4 Weeks**									
				**S1-C**	**S2-P**	**S3-Syn1**	**S4-Syn1.5**	**S5-Syn2**					
**Ethanol**	64-17-5	932	RI, MS	1.72 ± 0.66	59.89 ± 4.33	306.81 ± 22.08	304.44 ± 18.20	207.97 ± 13.63					
**3-Hydroxy butanal**	107-89-1	1009	MS	n.d	n.d	49.10 ± 2.22	n.d	n.d					
**2-Methyl-1-propanol**	78-83-1	1092	RI, MS	n.d	n.d	n.d	52.58 ± 6.17	39.24 ± 2.66					
**1-Butanol-3-methyl-acetate**	123-92-2	1122	RI, MS	n.d	9.21 ± 1.21	19.64 ± 1.67	5.98 ± 0.87	n.d					
**Limonene**	138-86-3	1200	RI, MS	n.d	79.60 ± 5.08	162.28 ± 7.30	58.53 ± 3.22	n.d					
**L-Limonene**	5989-54-8	1204	RI, MS	n.d	n.d	n.d	66.89 ± 4.28	100.85 ± 7.37					
**3-Methyl-1-butanol,**	123-51-3	1209	RI, MS	n.d	247.39 ± 16.12	671.56 ± 27.03	736.08 ± 25.31	526.66 ± 16.03					
**Butanoic acid-3-methyl butyl ester**	106-27-4	1259	RI, MS	n.d	n.d	n.d	n.d	n.d					
**Acetoin**	513-86-0	1284	RI, MS	36.16 ± 3.09	145.50 ± 9.33	199.63 ± 6.22	117.73 ± 7.28	82.85 ± 4.66					
**Octanoic acid ethyl ester**	106-32-1	1435	RI, MS	n.d	n.d	n.d	32.22 ± 4.11	23.15 ± 2.61					
**Acetic acid**	64-19-7	1449	RI, MS	n.d	22.13 ± 3.12	100.12 ± 3.54	45.57 ± 3.48	40.73 ± 5.04					
**2-Methyl-propanoic acid**	79-31-2	1570	RI, MS	n.d	n.d	35.11 ± 4.11	16.22 ± 1.78	14.84 ± 2.11					
**Butanoic acid**	107-92-6	1625	RI, MS	n.d	n.d	21.78 ± 2.33	n.d	6.59 ± 1.03					
**2-Methyl-butanoic acid**	116-53-0	1662	RI, MS	n.d	n.d	n.d	n.d	n.d					
**Hexanoic acid**	142-62-1	1846	RI, MS	9.60 ± 0.91	28.01 ± 1.66	35.89 ± 5.14	49.81 ± 4.60	32.35 ± 2.55					
**Phenylethyl alcohol**	60-12-8	1906	RI, MS	n.d	n.d	n.d	24.14 ± 2.47	18.90 ± 1.72					
**Octanoic acid**	124-07-2	2060	RI, MS	14.37 ± 1.40	28.43 ± 2.32	37.63 ± 3.55	39.89 ± 3.04	33.56 ± 2.36					

^a^ The retention index (RI) was determined in DB-WAX column via injection of a mix of *n*-alkane (c6–c23); S1-C: control plain yogurt; S2-P: yogurt with *S. boulardii*; S3-Syn1: yogurt with *S. boulardii* + 1% inulin; S4-Syn1.5: yogurt with *S. boulardii* + 1.5% inulin; S5-Syn2: yogurt with *S. boulardii* + 2% inulin; n.d: not detected.

## Data Availability

Not applicable.

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
