# Peer review of "Effect of Chilled Storage on Antioxidant Capacities and Volatile Flavors of Synbiotic Yogurt Made with Probiotic Yeast Saccharomyces boulardii CNCM I-745 in Combination with Inulin"

_jof, 2022, doi:10.3390/jof8070713_

Round 1

Reviewer 1 Report

Referees Comments

on the manuscript entitled “Effect of chilled storage on antioxidant capacities and volatile

flavors of synbiotic yogurt made with probiotic yeast Saccharomyces boulardii CNCM I-745 in combination with inulin” for Journal of Fungi

The authors studied the effect of chilled storage on antioxidant capacities and volatile flavors of synbiotic yogurt made in combination the probiotic yeast Saccharomyces boulardii and the inulin. The work was carried out using a wide range of modern biochemical methods. The results obtained are described and discussed in sufficient detail. The article may be published in the Journal of Fungi, however the minor corrections to the manuscript are required.

Point 1: Abstract, line 35: Deciphering the abbreviation DPPH.

Point 2, Page 4, line 139: Deciphering the abbreviation TPC.

Point 3, Results and Discussion: The titles of the subsections in “Results and Discussion” duplicate the titles of the methods. It's better to change them.

Point 3, Page 6: Make the design of table 2 and other tables in the traditional manner. I propose to change the title of Table 2 to “Influence of storage time on the antioxidant activity of different Yogurt samples”.

Author Response

Ms. Lina Zhang
Assistant Editor

Journal of Fungi,

Dear Ms Zhang,

First, we would like to thank the respected reviewers for giving time to our article and giving their valuable feedback.

Please find the attached response to reviewer comments on our research article manuscript entitled “Effect of chilled storage on antioxidant capacities and volatile flavors of synbiotic yogurt made with probiotic yeast Saccharomyces boulardii CNCM I-745 in combination with inulin” with manuscript id” “jof-1790333”

Reviewer 1

Referees Comments

on the manuscript entitled “Effect of chilled storage on antioxidant capacities and volatile

flavors of synbiotic yogurt made with probiotic yeast Saccharomyces boulardii CNCM I-745 in combination with inulin” for Journal of Fungi

The authors studied the effect of chilled storage on antioxidant capacities and volatile flavors of synbiotic yogurt made in combination the probiotic yeast Saccharomyces boulardii and the inulin. The work was carried out using a wide range of modern biochemical methods. The results obtained are described and discussed in sufficient detail. The article may be published in the Journal of Fungi, however the minor corrections to the manuscript are required.

Point 1: Abstract, line 35: Deciphering the abbreviation DPPH.

AR: Thank you very much for your comment. The abbreviation of DDPH has been added as DPPH (2,2-diphenyl-1-picrylhydrazyl) and highlighted in the revised manuscript. Please see revised manuscript.

Point 2, Page 4, line 139: Deciphering the abbreviation TPC.

AR: Thank you very much for your comment. TPC (Total phenol contents) has been added to the revised manuscript and highlighted in red color. Please see revised manuscript.

Point 3, Results and Discussion: The titles of the subsections in “Results and Discussion” duplicate the titles of the methods. It's better to change them.

AR: Thank you very much for your comment. These titles have been changed accordingly. Please see revised manuscript.

Point 3, Page 6: Make the design of table 2 and other tables in the traditional manner. I propose to change the title of Table 2 to “Influence of storage time on the antioxidant activity of different Yogurt samples”.

AR: Thank you very much for your comment. Changes have been made in the revised manuscript accordingly and Title of Table 2 has been changed to “Influence of storage time on the antioxidant activity of different Yogurt samples” and highlighted in red color. Please see revised manuscript.

Thank you once again.

Regards

Dr Yang Zhennai (PhD, Postdoc)

Professor

Beijing Advance Innovation Center for Food Nutrition and Human Health

Beijing Technology & Business University, Haidian Beijing, China

Email: yangzhennai@163.com

Reviewer 2 Report

Introduction may include some references related to naturally dairy products containing Lactic acid bacteria and yeast, like kefir type products.

Line 72: Why “also”? You compare with what?

Line 76: introduced where?

Line 117: replace “with” with “by”

Line 183: quantitation or quantification?

Line 250: Table 3 title is missing

Line 260: what do you mean when using both words “influenced” and “affected”?4

Line 301: can you provide any comment related to the ethanol content? Is it comparable with the kefir type products? Is the final product a “low alcohol” product?

Subsection 3.2 supports larger discussions in relation to the contribution to all the detected substances to the final organoleptic profile of the product

Subsection 3.3 supports more explanation on the common characteristics of the samples in each of the identified group

Author Response

Ms. Lina Zhang
Assistant Editor

Journal of Fungi,

Dear Ms Zhang,

First, we would like to thank the respected reviewers for giving time to our article and giving their valuable feedback.

Please find the attached response to reviewer comments on our research article manuscript entitled “Effect of chilled storage on antioxidant capacities and volatile flavors of synbiotic yogurt made with probiotic yeast Saccharomyces boulardii CNCM I-745 in combination with inulin” with manuscript id” “jof-1790333”

Reviewer 2

Introduction may include some references related to naturally dairy products containing Lactic acid bacteria and yeast, like kefir type products.

AR: Thank you very much for you comment. A Para has been added in the introduction part and highlighted in red color. Please see revised manuscript.

Line 72: Why “also”? You compare with what?

AR: Thank you very much for you comment. Compare with other dairy products these lines were added in the manuscript and highlighted (cheese, various lactic acid bacteria drinks, mixture of probiotic (fermented) milks and several types of yogurts including synbiotic yogurt containing both probiotics and prebiotics

Line 76: introduced where?

AR: Thank you very much for you comment. Introduced in Added in animal models in vivo. It has been added to the revised manuscript and highlighted in red color. Please see revised manuscript.

Line 117: replace “with” with “by”

AR: Thank you very much for you comment. It has been changed accordingly and highlighted in red color. Please see revised manuscript.

Line 183: quantitation or quantification?

AR: Thank you very much for you comment. It is Quantitation.

Line 250: Table 3 title is missing

AR: Thank you very much for you comment. The title of Table 3 has been added to the revised manuscript and highlighted in red color. Please see revised manuscript.  

Line 260: what do you mean when using both words “influenced” and “affected”?4

AR: Thank you very much for you comment. The storage condition has influenced the volatiles concentration in in terms synbiotic yogurt samples while has affected in case of control and probiotic yogurt that was reduced with the storage. To reduce the ambiguity affected word has been removed from the revised manuscript. Please see revised manuscript.

Line 301: can you provide any comment related to the ethanol content? Is it comparable with the kefir type products? Is the final product a “low alcohol” product?

AR: Thank you very much for you comment. Yes, it is a low alcohol product as compared to kefir. In terms of the final fermentation products of kefir lactic acid and ethanol were the main ones, whereas only traces of acetic acid were detected. The concentration of ethanol was ~0.9% w/v (~9 g/L) as reported by Tzavaras 2022, Similarly chen et al 2020 also reported that ethanol occupies one of the main components in volatiles as he studied the effect of lactic acid bacteria and yeasts on the structure and fermentation properties of Tibetan kefir grains.   

Subsection 3.2 supports larger discussions in relation to the contribution to all the detected substances to the final organoleptic profile of the product.

AR: Thank you very much for you comment. More discussion has been added to the revised manuscript and highlighted in red color. Please see revised manuscript.

Subsection 3.3 supports more explanation on the common characteristics of the samples in each of the identified group

AR: Thank you very much for you comment. More explanation has been added to the revised manuscript and highlighted in red color. Please see revised manuscript.

                                                Thank you once again.

Regards

Dr Yang Zhennai (PhD, Postdoc)

Professor

Beijing Advance Innovation Center for Food Nutrition and Human Health

Beijing Technology & Business University, Haidian Beijing, China

Email: yangzhennai@163.com

Round 2

Reviewer 2 Report

No further comments